# The evolution of a new cell type was associated with competition for a signaling ligand

Charles A. Ettensohn *, Ashrifia Adomako-Ankomah

Department of Biological Sciences, Carnegie Mellon University, Pittsburgh, Pennsylvania, United States of America

* ettensohn@cmu.edu

**Data Availability Statement:** All relevant data are within the paper and its Supporting Information files.

## Abstract

There is presently a very limited understanding of the mechanisms that underlie the evolution of new cell types. The skeleton-forming primary mesenchyme cells (PMCs) of euechinoid sea urchins, derived from the micromeres of the 16-cell embryo, are an example of a recently evolved cell type. All adult echinoderms have a calcite-based endoskeleton, a synapomorphy of the Ambulacraria. Only euechinoids have a micromere-PMC lineage, however, which evolved through the co-option of the adult skeletogenic program into the embryo. During normal development, PMCs alone secrete the embryonic skeleton. Other mesoderm cells, known as blastocoelar cells (BCs), have the potential to produce a skeleton, but a PMC-derived signal ordinarily prevents these cells from expressing a skeletogenic fate and directs them into an alternative developmental pathway. Recently, it was shown that vascular endothelial growth factor (VEGF) signaling plays an important role in PMC differentiation and is part of a conserved program of skeletogenesis among echinoderms. Here, we report that VEGF signaling, acting through ectoderm-derived VEGF3 and its cognate receptor, VEGF receptor (VEGFR)-10-Ig, is also essential for the deployment of the skeletogenic program in BCs. This VEGF-dependent program includes the activation of *aristaless-like homeobox 1* (*alx1*), a conserved transcriptional regulator of skeletogenic specification across echinoderms and an example of a "terminal selector" gene that controls cell identity. We show that PMCs control BC fate by sequestering VEGF3, thereby preventing activation of *alx1* and the downstream skeletogenic network in BCs. Our findings provide an example of the regulation of early embryonic cell fates by direct competition for a secreted signaling ligand, a developmental mechanism that has not been widely recognized. Moreover, they reveal that a novel cell type evolved by outcompeting other embryonic cell lineages for an essential signaling ligand that regulates the expression of a gene controlling cell identity.

**Funding:** This work was supported by a National Science Foundation Grant IOS-1354973 (CAE). The funders had no role in study design, data collection and analysis, decision to publish, or preparation of the manuscript.

**Competing interests:** The authors have declared that no competing interests exist.

**Abbreviations:** *alx1, aristaless-like homeobox 1*; BC, blastocoelar cell; ChIP-seq, chromatin immunoprecipitation sequencing; DIC, differential interference contrast microscopy; Dpp, Decapentaplegic; *erg, ets-related gene*; Ets1, E26 transformation-specific 1; *gatac, GATA-binding factor c*; GRN, gene regulatory network; hpf, hours postfertilization; *Lv-alx1, L. variegatus alx1*; *Lv-tbr, L. variegatus tbr*, *L. variegatus vegfr-10-Ig, L. variegatus vegfr-10-Ig*; mAb, monoclonal antibody; MO, morpholino; *msp130rel2, L. variegatus mesenchyme specific protein 130-related 2*; *pmar1, paired-class micromere anti-repressor*; PMC, primary mesenchyme cell; POL, polarized light microscopy; QPCR, quantitative PCR; RITC, rhodamine isothiocyanate; RNA-seq, RNA sequencing; RT-PCR, reverse transcription PCR; *scl, stem cell leukemia*; VEGF, vascular endothelial growth factor; VEGFR, VEGF receptor; WGA, wheat germ agglutinin; WMISH, whole-mount in situ hybridization.

# Introduction

Although all adult echinoderms possess calcite-based endoskeletal elements, only euechinoid sea urchins form micromeres and primary mesenchyme cells (PMCs). This newly evolved cell type is rigidly specified early in development through the activity of localized maternal factors and deploys a well-characterized gene regulatory network (GRN) that controls the morphogenetic behaviors and biomineral-forming properties of PMCs [1–3]. Much evidence suggests that the evolutionary appearance of this mesodermal cell lineage was associated with the co-option of an ancestral, adult program of skeletogenesis into the early embryo [3–5]. The regulation of skeletogenesis by vascular endothelial growth factor (VEGF) signaling was likely a component of this ancestral program [4,6–8]. In euechinoid sea urchins, although VEGF signaling is not involved in the maternally entrained, cell-autonomous specification of the micromere-PMC lineage, it plays an important role later in embryogenesis when PMC migration and skeletogenesis come under the regulatory influence of VEGF3 produced by ectoderm cells [9–12]. Morgulis and coworkers [13] recently used whole-embryo RNA sequencing (RNA-seq) to show that VEGF signaling regulates hundreds of genes, including many biomineralization genes expressed selectively by PMCs, and they propose that VEGF regulates an ancient program of tubulogenesis shared by echinoderms and vertebrates.

In contrast to PMCs, which are rigidly committed to a single (skeletogenic) fate, blastocoelar cells (BCs) are multipotent. These cells ordinarily give rise to a heterogeneous population of migratory, immunocyte-like cells [14]. They also possess skeletogenic potential, but during normal development, a signal from PMCs suppresses this potential and directs BCs to express an immunocyte fate (Fig 1). If PMCs are ablated at or about the time of ingression, BCs

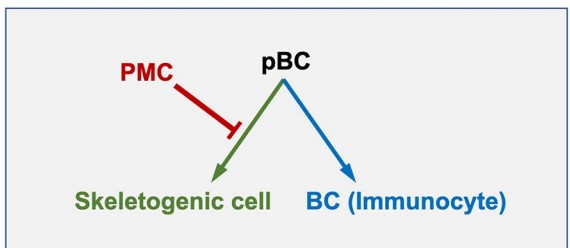

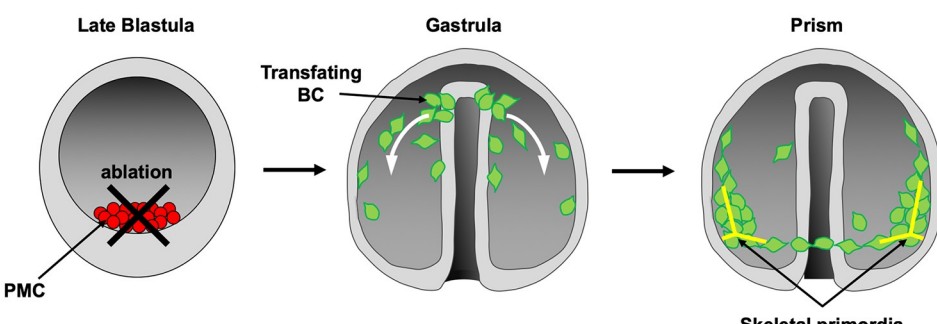

**Fig 1. PMC-derived signals suppress the skeletogenic potential of BCs.** Upper panel: Each pBC is multipotent and can adopt either a skeletogenic (PMC-like) or BC (immunocyte) fate. Ordinarily, a signal from PMCs (red bar) suppresses the skeletogenic potential of pBCs and causes them to adopt an immunocyte fate. Lower panels: Ablation of PMCs (red cells) at the late (mesenchyme) blastula stage leads to transfating of BCs (green cells). In these PMC(−) embryos, BCs ingress late in gastrulation but migrate to PMC-specific target sites on the blastocoel wall (indicated by white arrows), fuse to form a syncytium, and synthesize a correctly patterned skeleton, beginning with the formation of two triradiate skeletal primordia (shown in yellow). BC, blastocoelar cell; pBC, presumptive BC; PMC, primary mesenchyme cell.

undergo a striking change in phenotype; they adopt PMC-specific morphogenetic behaviors and secrete a correctly patterned skeleton [15–18]. This change in cell phenotype is associated with the molecular reprogramming of BCs, which ectopically deploy the skeletogenic GRN while extinguishing the expression of two regulatory genes, *stem cell leukemia* (*scl*) and *GATA-binding factor c* (*gatac*), that are associated with the immunocyte fate [14,19]. Although this cellular interaction was first described more than 50 years ago [20], the molecular nature of the PMC-derived signal has not been determined.

## Results

### VEGFR signaling is required for the activation and maintenance of the skeletogenic network in BCs

There are two VEGF receptor (VEGFR) genes in sea urchins: *vegfr-10-Ig* is ordinarily expressed at high levels (peak expression > 3,000 transcripts/embryo), whereas *vegfr-7-Ig* is expressed at much lower levels (<150 transcripts/embryo) [21,22]. During normal development, *vegfr-10-Ig* is expressed selectively by presumptive PMCs beginning at the early blastula stage [9]. In PMC(−) embryos, however, *vegfr-10-Ig* is expressed robustly in BCs [19]. The spatial expression pattern of *vegfr-7-Ig* is not known in detail but, unlike *vegfr-10-Ig*, this gene is not differentially expressed by PMCs [23,24].

Axitinib is a highly selective inhibitor of VEGF receptors at nanomolar concentrations [25,26]. The effects of axitinib on sea urchin development mimic those of VEGF3 and VEGFR-10-Ig knockdowns, confirming that axitinib is a specific inhibitor of VEGF/VEGFR signaling in this system [9,11]. We found that PMC(−) *Lytechinus variegatus* embryos treated continuously with axitinib (75 nM) beginning at the time of PMC removal failed to form a skeleton, even after prolonged culture (48 hours postfertilization [hpf]) (Fig 2A and 2A'). Axitinib-treated embryos swam vigorously and gastrulated at the same time as untreated, PMC(−) embryos but never extended arms. Polarization microscopy showed that these embryos completely lacked birefringent skeletal elements (Fig 2B and 2B'). This phenotype was reproducible and highly penetrant; >95% of axitinib-treated, PMC(−) embryos showed this effect across many batches.

Early molecular events in BC transfating include the activation of *aristaless-like homeobox 1* (*alx1*) and *t-brain* (*tbr*), two regulatory (i.e., transcription factor-encoding) genes that function early in the PMC GRN and are ordinarily expressed only in the large micromere-PMC lineage [17,19]. Alx1 is a critically important regulator of the skeletogenic GRN both in the PMC lineage and in transfating BCs, whereas *tbr* plays a less prominent role [17,24,27–30]. The activation of these two genes in transfating BCs is detectable by whole-mount in situ hybridization (WMISH) as early as 2 hours post–PMC depletion [19]. We confirmed that *L. variegatus alx1* (*Lv-alx1*) and *L. variegatus tbr* (*Lv-tbr*) were expressed robustly in the anterior region of the archenteron in control PMC(−) embryos but found that expression of both genes was greatly reduced in embryos that were placed in axitinib at the time of PMC removal (Fig 2C–2D') (3 trials, *n* = 15–30 embryos/trial). This effect was readily apparent by 3 hours post–PMC depletion when assayed by conventional (histochemical) WMISH. These findings demonstrated that the expression of *alx1* and *tbr* in BCs was dependent upon VEGF/VEGFR signaling. Consistent with this finding, the expression of two terminal biomineralization genes, *p58a* and *L. variegatus mesenchyme specific protein 130-related 2* (*msp130rel2*), that receive positive regulatory inputs from *alx1* [24,31,32], was also greatly reduced in axitinib-treated, PMC(−) embryos (Fig 2E–2F', >90% of embryos, *n* > 40). We also immunostained embryos with monoclonal antibody (mAb) 6a9, a widely used marker of PMC differentiation that recognizes the biomineralization protein MSP130 [15,33,34], and observed a dramatic reduction in the numbers of

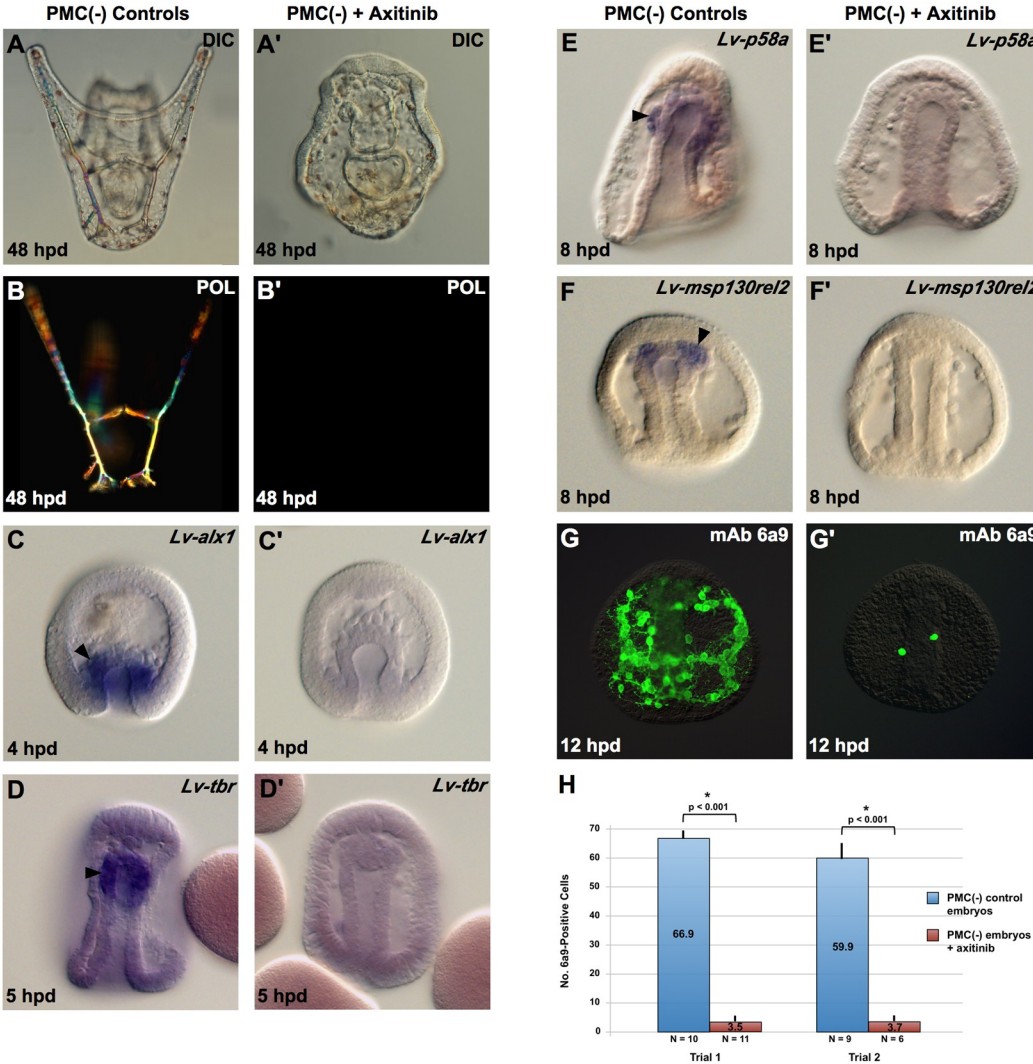

**Fig 2. Axitinib blocks BC transfating.** PMCs were removed from mesenchyme blastula–stage embryos, and the resultant PMC(−) embryos were separated into two cohorts. One cohort was left in plain seawater, whereas the other was transferred to 5 nM axitinib immediately after PMC removal. (A-G) Control PMC(−) embryos. (A'-G') Axitinib-treated PMC(−) embryos. Axitinib treatment blocked the formation of birefringent skeletal elements (A-B') and the expression of early skeletogenic regulatory genes by BCs as shown by WMISH analysis of *Lv-alx1* (C,C') and *Lv-tbr* (D,D') expression. The expression of skeletogenic effector genes downstream of *alx1* was also blocked, as indicated by WMISH analysis of *Lv-p58a* (E,E') and *Lv-msp130rel2* (F,F'), and by immunostaining with mAb 6a9 (G,G'). Arrowheads indicate expression of skeletogenic genes by transfating BCs. Panel H shows quantification of 6a9-positive cells in control and axitinib-treated PMC(−) embryos at 12 hpd (two independent trials from separate matings). Statistical significance of the data was assessed by two-sided *t* tests, and *p*-values < 0.05 are indicated by asterisks. Raw data can be found in S1 Data. BC, blastocoelar cell; DIC, differential interference contrast microscopy; hpd, hours post–PMC depletion; *Lv-alx1, L. variegatus aristaless-like 1*; *Lv-tbr, L. variegatus t-brain*; mAb, monoclonal antibody; *msp130rel2, L. variegatus mesenchyme specific protein 130-related 2*; POL, polarized light microscopy; PMC, primary mesenchyme cell; WMISH, whole-mount in situ hybridization.

6a9(+) cells in axitinib-treated, PMC(−) embryos (Fig 2G and 2H). Culturing such embryos for an additional 24 hours did not increase the numbers of 6a9(+) cells, indicating that axitinib was not simply delaying BC transfating.

We next asked whether VEGFR signaling was required only at the onset of BC transfating (i.e., whether it acted like a trigger) or was required continuously during the transfating process. We removed PMCs from mesenchyme blastula–stage embryos and allowed them to

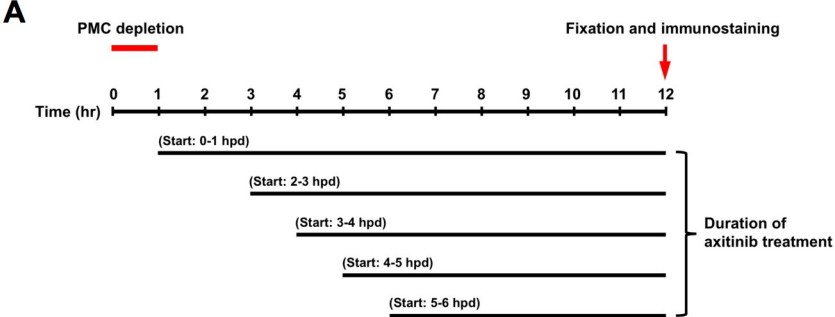

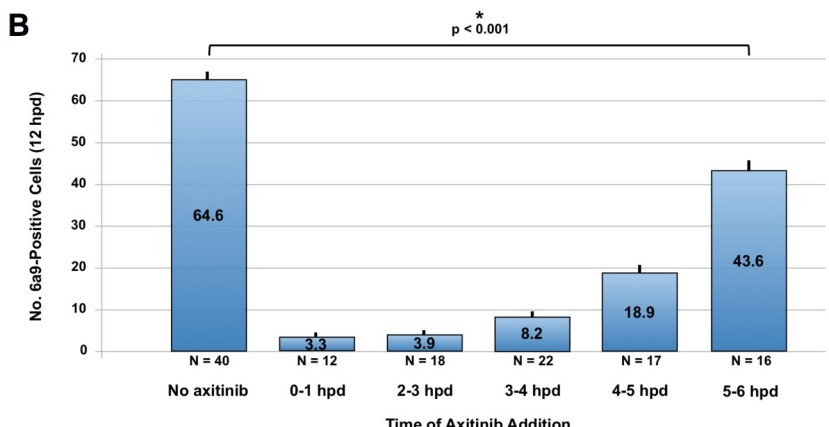

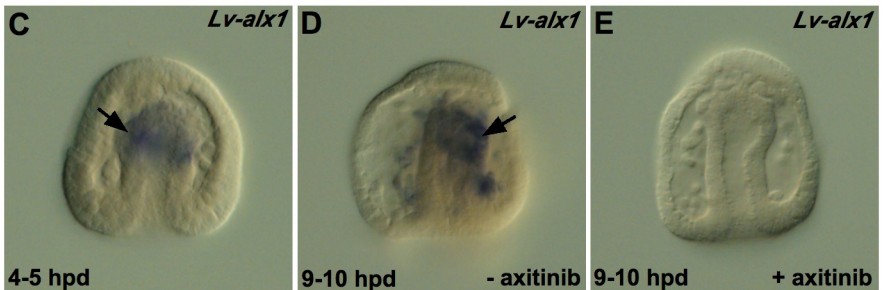

**Fig 3. Time course of axitinib sensitivity during BC transfating.** (A) Experimental design. Solid horizontal bars indicate the duration of axitinib (5 nM) treatment. Start times are shown as a 1-hour interval because microsurgical removal of PMCs required approximately 1 hour for each cohort of embryos. (B) Counts of 6a9-positive cells. Data shown were collected from several independent trials, and untreated controls from all trials were pooled. For all time periods tested, we observed a significant reduction in 6a9(+) cells in axitinib-treated PMC(−) embryos relative to untreated PMC(−) controls. Raw data can be found in S1 Data. (C-E) Signaling through VEGFR is required for the maintenance of *Lv-alx1* expression in BCs during reprogramming. PMC(−) embryos were treated with axitinib for 5 hours beginning at 4–5 hpd. At the start of axitinib treatment, *Lv-alx1* was expressed robustly by BCs at the tip of the archenteron (arrow in C, 10/11 embryos). Five hours later (9–10 hpd), *alx1* continued to be expressed in untreated PMC(−) embryos (arrow in D, 5/5 embryos), but expression was undetectable in most axitinib-treated PMC(−) embryos (E, 10/13 embryos). BC, blastocoelar cell; hpd, hours post–PMC depletion; *Lv-alx1*, *L. variegatus aristaless-like homeobox 1*; PMC, primary mesenchyme cell; VEGFR, vascular endothelial growth factor receptor.

develop for varying periods of time before adding axitinib to the medium (Fig 3A). At 12 hours post–PMC depletion, embryos were fixed and immunostained with mAb 6a9. As the time interval between PMC depletion and the addition of axitinib increased, the numbers of 6a9(+) cells increased progressively (Fig 3B). This suggested that VEGFR signaling was

required continuously during the first several hours after PMC depletion. Surprisingly, even when axitinib was added 5–6 hours post–PMC depletion, several hours after the activation of skeletogenic regulatory genes in BCs, we observed a significant reduction in the numbers of 6a9(+) cells. This finding raised the question of whether, once activated, early regulatory genes maintained their expression in the presence of axitinib. We removed PMCs at the mesenchyme blastula stage and allowed the resulting PMC(−) embryos to develop for 4–5 hours, sufficient time for *alx1* to be activated ectopically in BCs (Fig 3C). Transfer of PMC(−) embryos into 75 nM axitinib at that time led to a decline in *alx1* expression to undetectable levels in most embryos (10/13, or 77%) within 5 hours, whereas expression was maintained (or increased) in sibling, untreated, PMC(−) embryos (Fig 3D and 3E). These observations showed that VEGFR signaling was required not only to activate but also to maintain the expression of *alx1* in transfating BCs.

## VEGFR signaling acts through VEGFR-10-Ig and VEGF3

Because *vegfr-10-Ig* is expressed at high levels during embryogenesis and because it has been shown to regulate PMC skeletogenesis, it seemed likely that this receptor was the primary target of axitinib. Using WMISH, we first examined the expression of *L. variegatus vegfr-10-Ig* (*Lv-vegfr-10-Ig*) in control embryos. We confirmed that *vegfr-10-Ig* was expressed robustly in PMCs, as previously reported [9], but also observed faint expression in the wall of the archenteron at the early gastrula stage (S1 Fig). This region represents the presumptive nonskeletogenic mesoderm, including the presumptive BCs. Expression in the nonskeletogenic mesoderm was observed in multiple batches of embryos; however, this expression was apparent in only 30%–40% of embryos in each batch and was not detectable at later developmental stages, suggesting that expression was highly transient. In PMC(−) embryos, we originally documented high levels of expression of *vegfr-10-Ig* in transfating BCs at a relatively late stage, 10–11 hours post–PMC depletion, several hours after *alx1* activation [19]. Cheng and coworkers [35], however, used quantitative PCR (QPCR) and WMISH to show that in micromere(−) embryos, which resemble PMC(−) embryos in many respects, expression of *vegfr-10-Ig* and *alx1* was up-regulated at almost the same time. We therefore reexamined the timing of *vegfr-10-Ig* activation in PMC(−) embryos. We found that within 2 hours after PMC removal, *vegfr-10-Ig* mRNA was expressed robustly in the vegetal region of >90% of PMC(−) embryos, in an area that included the BC domain (Fig 4A and 4B) (2 trials, 10–20 embryos/trial). Expression remained strong at 7 hours post–PMC depletion (Fig 4C). The timing of *vegfr-10-Ig* up-regulation in BCs following PMC removal therefore closely paralleled the timing of *alx1* and *tbr* activation, in agreement with the findings of Cheng and coworkers [35].

The rapidity with which *vegfr-10-Ig* mRNA accumulated after PMC removal argued against the possibility that activation occurred via a relay mechanism that involved the transcriptional activation of an intermediate regulatory gene. Such a relay mechanism would require new protein synthesis in order to produce the relevant regulatory factor. To explore this possibility, we treated embryos with 100 μM emetine, which rapidly (<25 minutes) blocks >95% of protein synthesis in *L. variegatus* embryos [36]. Microsurgery was carried out in the presence of the inhibitor, and PMC(−) embryos were allowed to develop for an additional 2 hours in emetine-containing seawater before they were processed for WMISH. Longer incubation times were not possible, as embryos disaggregated when exposed to emetine for longer than approximately 2.5 hours. We observed a strong activation of *vegfr-10-Ig* in >90% of emetine-treated PMC(−) embryos (3 trials, 10–20 embryos/trial) (Fig 4D), arguing against a transcriptional relay mechanism.

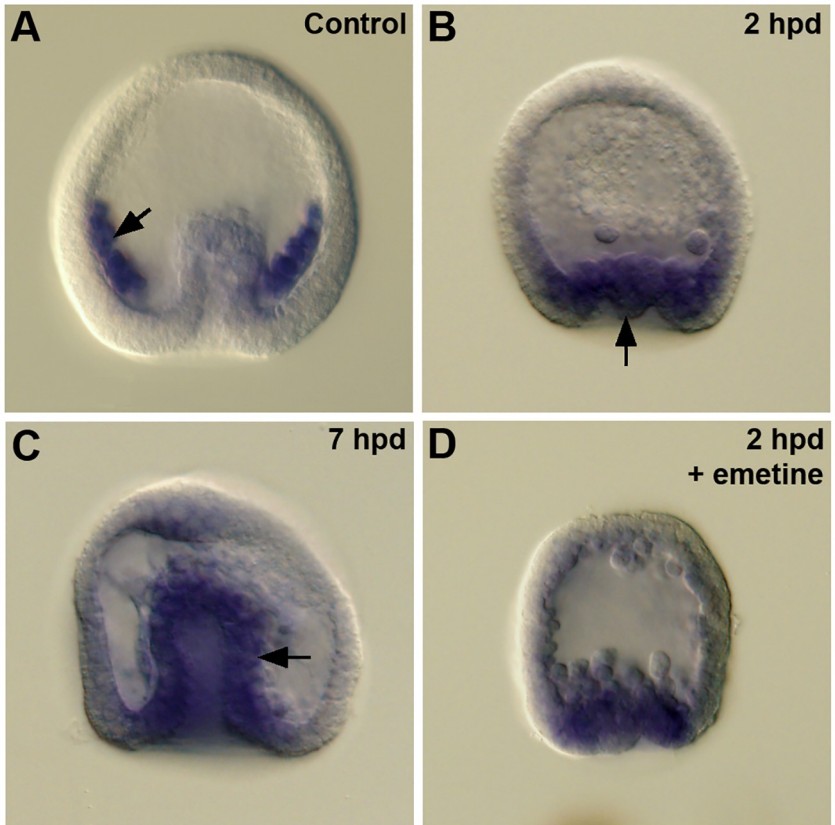

**Fig 4. *Lv-vegfr-10-Ig* expression during BC transfating.** (A) Control gastrula. *Lv-vegfr-10-Ig* is expressed at high levels by PMCs (arrow), as previously reported [9]. (B) PMC(−) embryo, 2 hpd. Expression of *Lv-vegfr-10-Ig* is evident in the invaginating vegetal plate (arrow), a region that includes presumptive BCs (32/37 embryos). (C) PMC(−) embryo, 7 hpd. Expression of Lv-vegfr-10-Ig is apparent in the wall of the archenteron (arrow). (D) Emetine-treated PMC(−) embryo, 2 hpd. Inhibition of protein synthesis did not prevent the robust expression of *Lv-vegfr-10-Ig* in the vegetal plate following PMC removal (16/20 embryos). BC, blastocoelar cell; hpd, hours post–PMC depletion; *Lv-vegfr-10-Ig*, *L. variegatus vascular endothelial growth factor receptor-10-Ig*; PMC, primary mesenchyme cell.

To test whether signaling through VEGFR-10-Ig was required for BC transfating, we inhibited the expression of the receptor using a splice-blocking morpholino (MO), following the strategy of Duloquin and coworkers [9] (Fig 5A). Embryos injected with the *Lv-vegfr-10-Ig* MO showed reduced levels of correctly spliced, wild-type mRNA and expressed high levels of an alternatively spliced RNA species that contained intron 2 (Fig 5B). We cloned and sequenced the prominent 1.1-kb PCR product shown in Fig 5B and confirmed that intron 2 was present, resulting in the introduction of multiple stop codons in all three reading frames. Less abundant PCR products were also detectable (asterisks in Fig 5B), which may reflect utilization of cryptic splice sites within exon 2 and/or intron 2. We concluded that the splice-blocking MO resulted in a marked, but incomplete, knockdown of VEGFR-10-Ig.

Knockdown of VEGFR-10-Ig in PMC(−) embryos produced a phenotype indistinguishable from axitinib treatment. Morphant PMC(−) embryos gastrulated normally and developed a tripartite gut, ciliary band, pigment cells, and coelomic pouches but lacked birefringent skeletal elements (Fig 5C–5D'). We also observed a striking decrease in the number 6a9(+) cells in PMC(−) morphant embryos (Fig 5E and 5F). This effect was not due to a delay in transfating, as shown by immunostaining morphant embryos at later time points (20 hours post–PMC

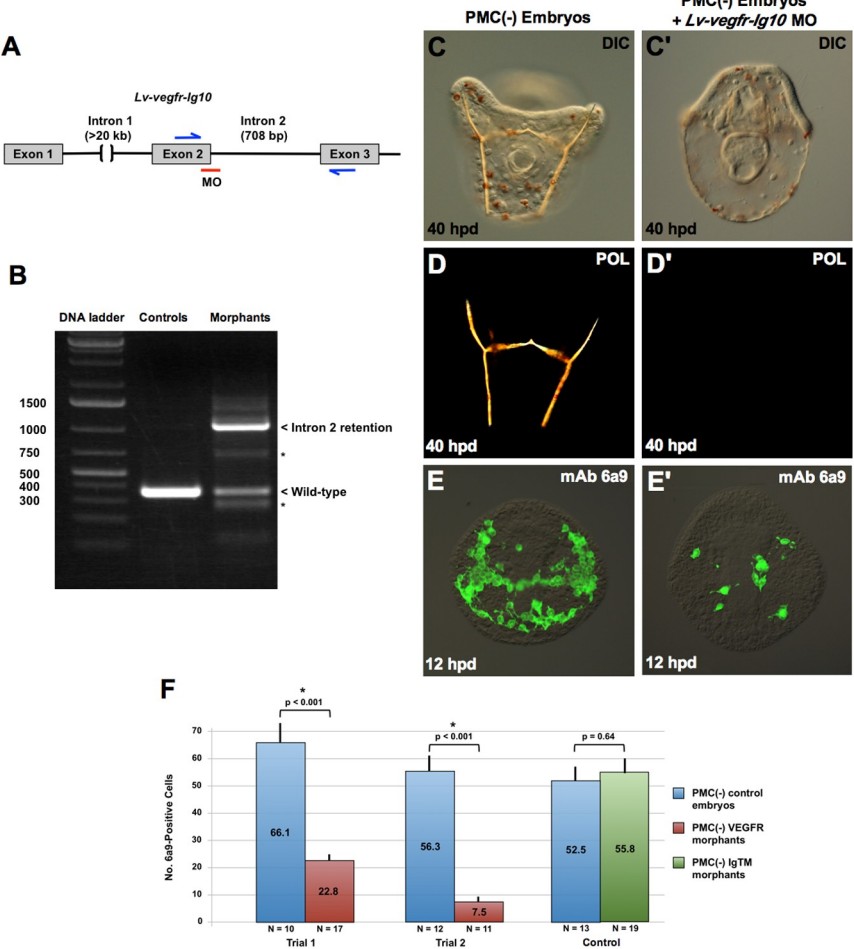

**Fig 5. Lv-VEGFR-10-Ig is required for BC transfating.** (A) MO knockdown strategy. Following the approach of Duloquin and coworkers [9], an MO was directed at the exon 2/intron 2 splice junction of the *Lv-vegfr-10-Ig* primary transcript. (B) RT-PCR analysis of morphant embryos. The *Lv-vegfr-10-Ig* MO produced a significant reduction in the level of wild-type mRNA and the appearance of a prominent splicing isoform that included intron 2, as verified by cloning and sequencing of the PCR product. Inclusion of intron 2 resulted in multiple stop codons in all reading frames and the production of a truncated, nonfunctional receptor. Low levels of other mis-spliced forms of *Lv-vegfr-10-Ig* mRNA (asterisks) were also detected. (C-D) BC transfating and skeletogenesis were suppressed in PMC(−), morphant embryos, as revealed by DIC and polarization microscopy at 40 hpd (C-D') and by immunostaining with mAb 6a9 at 12 hpd (E-F). BC reprogramming was not affected by an equivalent concentration of an MO directed against Lv-IgTM, a PMC-specific protein that regulates skeletal branching [37]. Statistical significance of the data was assessed by two-sided *t* tests and *p*-values < 0.05 are indicated by asterisks. Raw data can be found in S1 Data. BC, blastocoelar cell; DIC, differential interference contrast microscopy; hpd, hours post–PMC depletion; *Lv-vegfr-10-Ig*, *L. variegatus vascular endothelial growth factor receptor-10-Ig*; Lv-IgTM, *L. variegatus* Ig and transmembrane domain protein; mAb, monoclonal antibody; MO, morpholino; PMC, primary mesenchyme cell; POL, polarized light microscopy; RT-PCR, reverse transcription PCR.

depletion). In addition, the effect was specific to the VEGFR MO, as injection of an equivalent concentration of an MO complementary to IgTM, a protein that regulates skeletal branching [37], had no effect on BC transfating (Fig 5F). Previous work showed that *delta* MO also had no effect of BC transfating [19]. These studies showed that signaling through VEGFR-10-Ig was required for BCs to express a skeletogenic fate. They suggested that axitinib blocked transfating by inhibiting the function of this receptor in BCs, where the gene is expressed robustly following PMC removal.

Three genes in the sea urchin genome—*vegf*, *vegf2*, and *vegf3*—encode ligands in the VEGF family [21]. During gastrulation, *vegf3* is expressed at high levels (peak expression > 1,200 transcripts/embryo), whereas *vegf* and *vegf2* are expressed at very low levels or not at all (<20 transcripts/embryo) [22]. *vegf3* is expressed in localized regions of the ectoderm and functions along with *vegfr-10-Ig* as an essential mediator of PMC migration and skeletogenesis [9–13]. VEGF3 was therefore a likely candidate for the ligand that interacted with VEGFR-10-Ig to activate the skeletogenic GRN in transfating BCs. To test this hypothesis, we knocked down VEGF3 expression using a translation-blocking MO characterized previously [11]. Knock-down of VEGF3 in PMC(−) embryos produced a phenotype indistinguishable from VEGFR-10-Ig knockdown and axitinib treatment. PMC(−) morphant embryos gastrulated on schedule and gave rise to normal, prism-like larvae that lacked arms and birefringent skeletal elements (Fig 6A–6B'). These embryos also contained reduced numbers of 6a9(+) cells (Fig 6C and 6D). Our observations demonstrated that VEGF3 (like VEGFR-10-Ig) was required for BC transfating and strongly supported the hypothesis that VEGF3 is the ligand that interacts with VEGFR-10-Ig.

## PMCs suppress the skeletogenic potential of BCs by sequestering VEGF3

The finding that signaling through the VEGF pathway, mediated by VEGF3 and VEGFR-10-Ig, was essential for the activation of the skeletogenic pathway in BCs suggested a possible mechanism for PMC-to-BC signaling. As noted above, PMCs ordinarily express high levels of *vegfr-10-Ig* during gastrulation. Our results suggested that PMCs might outcompete BCs for VEGF3, thereby preventing the activation of the skeletogenic program by BCs and directing them into an alternative (immunocyte) pathway.

This model generated two testable predictions. First, lowering the level of VEGFR-10-Ig on the PMC surface should reduce the ability of the cells to suppress BC transfating. We tested this hypothesis by generating chimeric embryos in which VEGFR-10-Ig expression was knocked down selectively in PMCs (Fig 7A–7E). The entire complement of PMCs was removed from host embryos, and varying numbers of labeled, donor PMCs were transplanted into the host embryos. PMCs were labeled by incubating donor embryos in a reactive dye, rhodamine isothiocyanate (RITC), which results in the covalent attachment of rhodamine to lysine residues on cell surface proteins [15]. The fluorescently tagged proteins do not readily diffuse from cell to cell after fusion; therefore, this method persistently labels the donor cells. Donor PMCs from control (uninjected) and VEGFR-10-Ig morphant embryos were tested separately. It was shown previously that the number of BCs that express a skeletogenic fate is inversely proportional to the number of PMCs in the blastocoel [15], a finding that we con-firmed (Fig 7F). Significantly, we observed that PMCs with reduced expression of VEGFR-10-Ig were much less effective than control PMCs in suppressing the skeletogenic potential of BCs (Fig 7F). The residual signaling activity of morphant PMCs was consistent with the incomplete nature of the VEGFR-10-Ig knockdown (Fig 5B). These experiments showed that signaling by PMCs was dependent upon their expression of VEGFR-10-Ig.

Both control and morphant PMCs migrated actively and accumulated at typical PMC target sites near the embryo equator; however, in about 50% of embryos, morphant PMCs remained localized predominantly at PMC target sites on one side of the blastocoel, in contrast to control PMCs, which formed a bilaterally symmetrical PMC ring pattern. This indicated that knock-down of VEGFR-10-Ig partially compromised PMC migration and/or patterning, as previously reported [9,11]. To test whether VEGFR-10-Ig MO affected signaling indirectly by perturbing PMC migration, we agglutinated PMCs by microinjecting wheat germ agglutinin (WGA) into the blastocoel at the mesenchyme blastula stage [38]. In most embryos, the PMCs

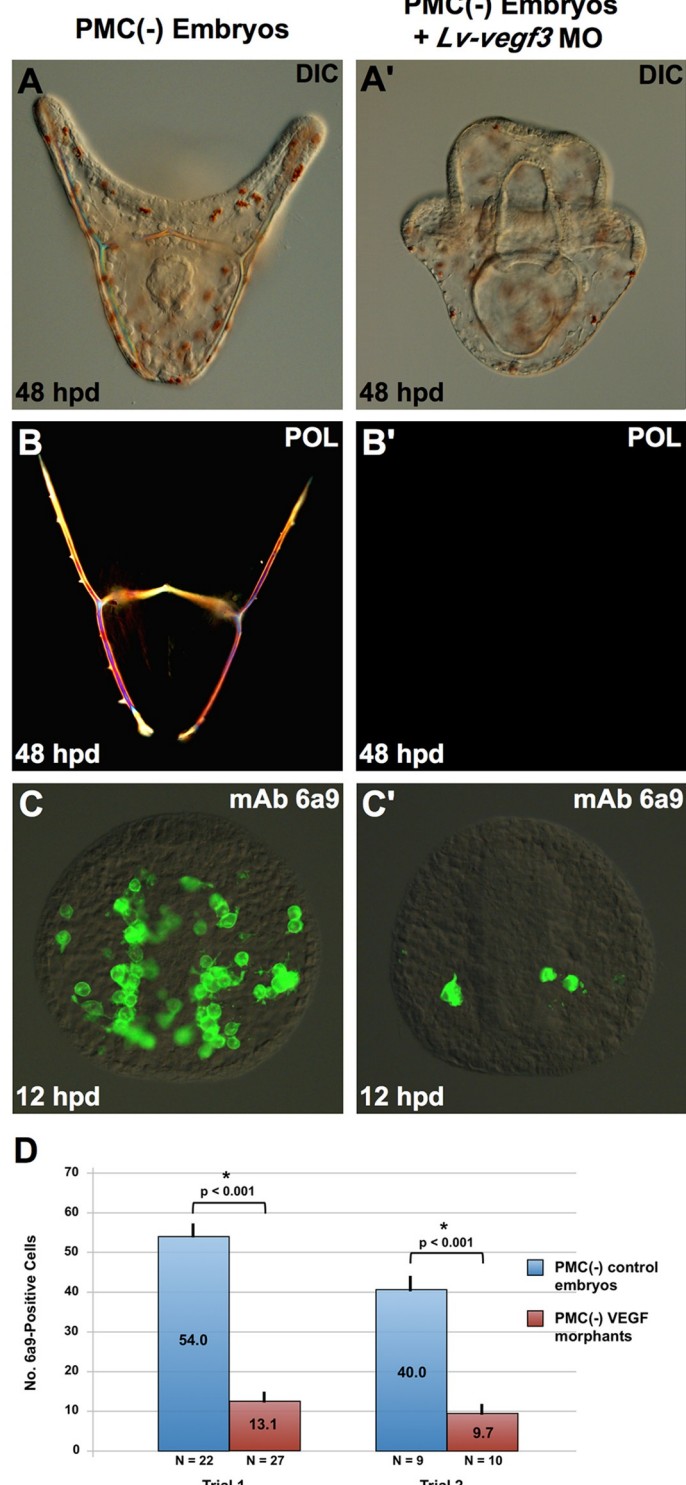

**Fig 6. Lv-VEGF3 is required for BC transfating.** A previously characterized translation-blocking MO was used to interfere with *Lv-vegf3* expression [11]. PMC(−) morphant embryos failed to form skeletal elements even at 48 hpd (A-B'), and very few 6a9(+) cells were detectable at 12 hpd (C-D), indicating that BC transfating was largely blocked. Statistical significance of cell count data was assessed by two-sided *t* tests, and *p*-values < 0.05 are indicated by asterisks. Raw data can be found in S1 Data. BC, blastocoelar cell; DIC, differential interference contrast microscopy; hpd, hours post–PMC depletion; *Lv-vegf3*, *L. variegatus vascular endothelial growth factor 3*; mAb, monoclonal antibody; MO, morpholino; PMC, primary mesenchyme cell; POL, polarized light microscopy.

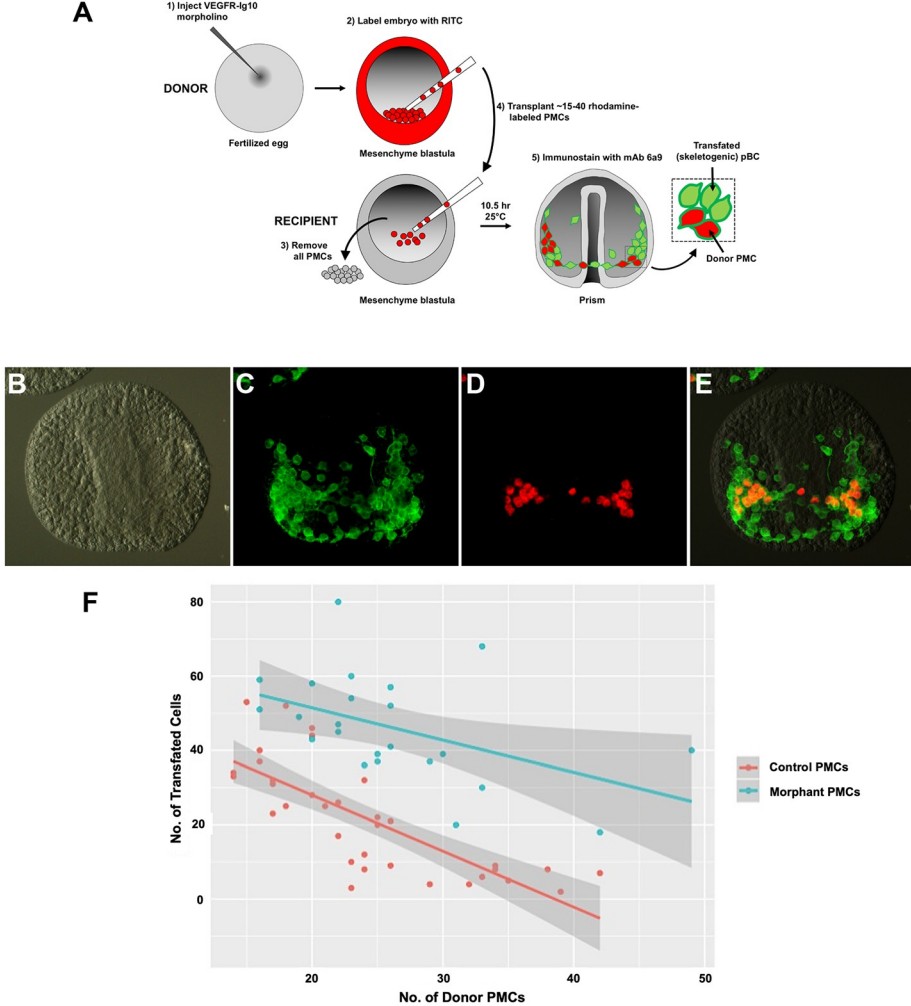

**Fig 7. Knockdown of Lv-VEGFR-10-Ig in PMCs reduces their ability to suppress BC transfating.** (A) Experimental design. The entire complement of PMCs was removed from a recipient embryo and replaced with 15–40 RITC-labeled PMCs from control or Lv-VEGFR-10-Ig morphant donor embryos. Then, 10.5 hours after transplantation, recipient embryos were fixed and immunostained with mAb 6a9 and a DyLight 488 goat anti-mouse secondary antibody. Donor PMCs were identified by (red + green) fluorescence, whereas transfated BCs exhibited only green fluorescence. (B-E) A representative embryo after immunostaining, viewed with DIC (B) or epifluorescence optics. (C) mAb 6a9 immunostaining showing all skeletogenic cells (donor PMCs and transfated BCs). (D) Rhodamine fluorescence showing donor PMCs. (E) Overlay of the two fluorescent channels. Cells that are green but not red are transfated BCs. (F) Quantification of the numbers of transfated BCs following PMC transplantation. Control PMCs (orange) were more potent at suppressing BC transfating than PMCs that had reduced expression of Lv-VEGFR-10-Ig (teal). In both cases, the number of transfated cells was inversely related to the number of PMCs in the blastocoel. Each point represents a single recipient embryo. Lines of best fit are indicated, and 95% confidence intervals are shown in gray. Raw data can be found in S2 Data. BC, blastocoelar cell; DIC, differential interference contrast microscopy; Lv-VEGFR-10-Ig, *L. variegatus* vascular endothelial growth factor receptor-10-Ig; mAb, monoclonal antibody; pBC, presumptive BC; PMC, primary mesenchyme cell; RITC, rhodamine isothiocyanate.

quickly (<2 hours) coalesced into 1–2 large masses that persisted for >12 hours (28/31 embryos) (S2 Fig). Nevertheless, PMCs were still capable of fully suppressing BC transfating, as shown by the absence of any 6a9(+) cells at the archenteron tip during gastrulation. Injection of WGA into PMC(−) embryos did not by itself prevent BC transfating, and 6a9(+) cells appeared normally (34/34 embryos). These studies showed that the reduced signaling potency

of morphant PMCs could not be attributed to the compromised migratory capacity of these cells.

A second prediction of the competition model was that overexpression of VEGF3 should saturate receptors on the PMC surface and provide sufficient unbound ligand to activate the skeletogenic program in BCs. During normal development, expression of VEGF3 is limited to small patches of ectoderm that overlie sites of skeletal growth. We overexpressed VEGF3 by microinjecting mRNA encoding full-length Lv-VEGF3 into fertilized eggs, causing the ligand to be expressed throughout the embryo. Injection of VEGF3 mRNA led to a concentration-dependent increase in the number of skeletogenic (6a9-positive) cells at the late prism stage (24 hpf) (Fig 8A, 8A' and 8D). The spatial patterning of 6a9(+) cells was markedly perturbed in embryos overexpressing VEGF3, consistent with the role of this molecule in PMC guidance [9]. Strikingly, when we examined *Lv-vegf3* mRNA–injected embryos several hours earlier in development, we observed ectopic activation of *Lv-alx1* (14/18 embryos, or 78%) and *Lv-tbr* (24/32 embryos, or 75%) in the wall of the archenteron, in a relatively broad territory that included presumptive mesoderm and possibly also presumptive endoderm cells (Fig 8B–8C'). Activation in presumptive endoderm is consistent with evidence that under appropriate experimental conditions, these cells can also adopt a skeletogenic fate, probably via the re-specification of a BC-like intermediate state [19,39]. *Lv-alx1* and *Lv-tbr* were not activated at the very anterior tip of the archenteron, the location of the small micromere descendants (future germ cells).

## Discussion

It has been proposed that receptor molecules might sequester signaling ligands, thereby influencing their effective range and tissue pattering [40]. In the *Drosophila* ovary, expression of the Decapentaplegic (Dpp) receptor by escort cells restricts Dpp distribution and its influence on the fates of cells within the stem cell niche [41]. Interactions between Wnt ligands and Frizzled receptors limit the range of Wnt movement in the mammalian intestinal stem cell niche [42]. Other studies have shown that manipulation of Hedgehog and Dpp receptor expression in *Drosophila* imaginal wing discs influences the range of signaling and subsequent tissue pattering [43,44]. The simplest hypothesis is that receptor molecules sequester ligands, thereby preventing them from reaching more distant cells, although indirect mechanisms may also operate [45]. How widespread such mechanisms are in controlling cell fates, particularly during early embryogenesis, remains an open question.

In the present study, we provide evidence that mesodermal cell fates during sea urchin gastrulation are regulated by direct competition for a paracrine factor, VEGF3. Two different groups of cells (PMCs and BCs) are capable of responding to the ligand, but as a result of competition, only one cell population does so, whereas the other is directed into an alternative developmental pathway (Fig 9). The following observations support the view that PMCs regulate BC fate by outcompeting these cells for VEGF3: (1) signaling through VEGFR-10-Ig, which is up-regulated in the presumptive BCs territory immediately following PMC ablation, is essential for the activation of *alx1* and the deployment of the skeletogenic GRN in BCs; (2) VEGF3, produced by the ectoderm, is also required for BCs to express a skeletogenic fate, presumably through the interaction of this ligand with VEGFR-10-Ig; (3) PMCs with reduced levels of *vegfr-10-Ig* have a reduced ability to suppress BC transfating; and (4) overexpression of VEGF3 is sufficient to override the suppressive influence of PMCs and induce BC transfating. It should be noted that injection of VEGF3 mRNA into fertilized eggs probably results in the secretion of the ligand by most or all cells of the embryo. Therefore, it is possible that the effects we see may be caused by VEGF3 produced by the BCs themselves acting in an autocrine fashion.

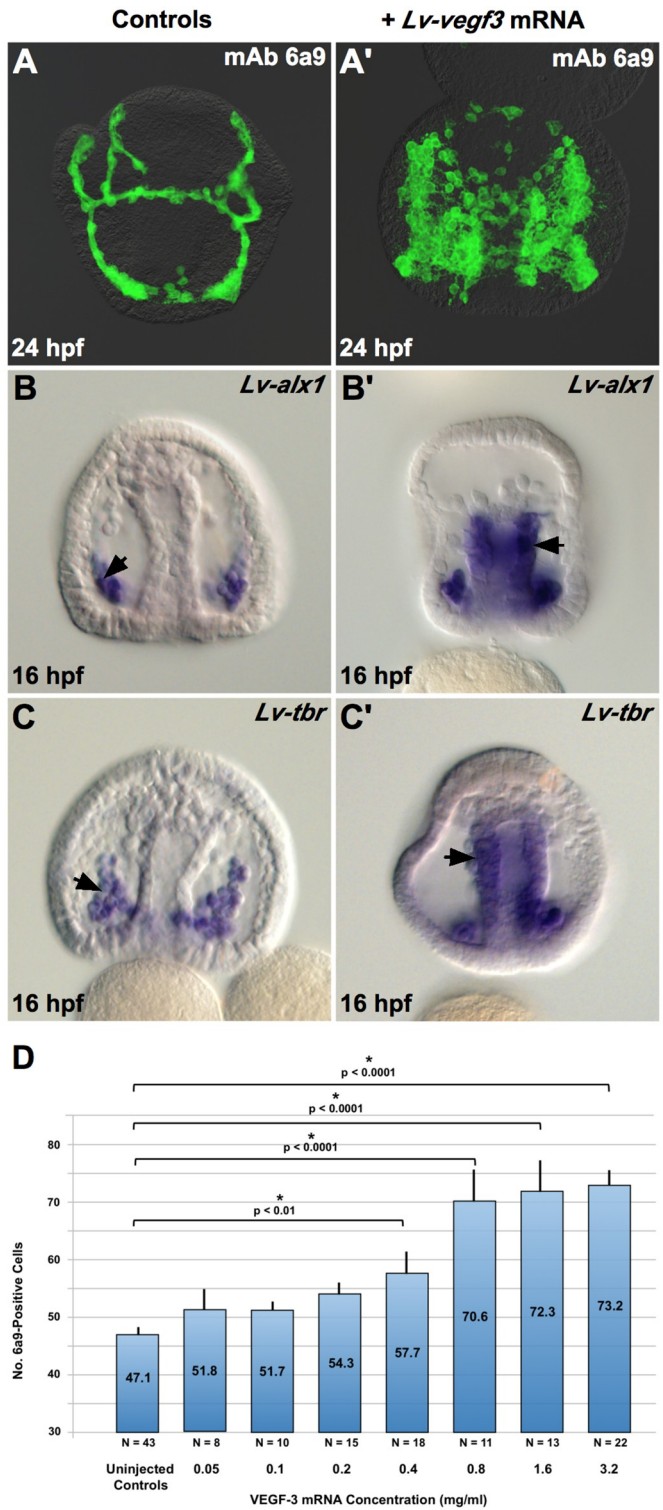

**Fig 8. Overexpression of Lv-VEGF3 induces BC transfating.** Capped mRNA encoding Lv-VEGF3 was injected into fertilized eggs. Many supernumerary 6a9(+) cells were observed in such embryos 24 hpf, when sibling control embryos had reached the late prism stage (A, A'). In control embryos, expression of *Lv-alx1* and *Lv-tbr* was restricted to PMCs, as expected (arrows in B and C), whereas injection of *Lv-vegf3* mRNA resulted in the ectopic activation of Lv-alx1 and Lv-tbr in the wall of the archenteron (arrows in B' and C'). The effect of *Lv-vegf3* mRNA was dose-dependent (D). Raw data can be found in S1 Data. BC, blastocoelar cell; hpf, hours postfertilization; Lv-alx1, *L. variegatus aristaless-like 1*; Lv-tbr, *L. variegatus t-brain*; Lv-VEGF3, *L. variegatus* vascular endothelial growth factor 3; mAb, monoclonal antibody; PMC, primary mesenchyme cell.

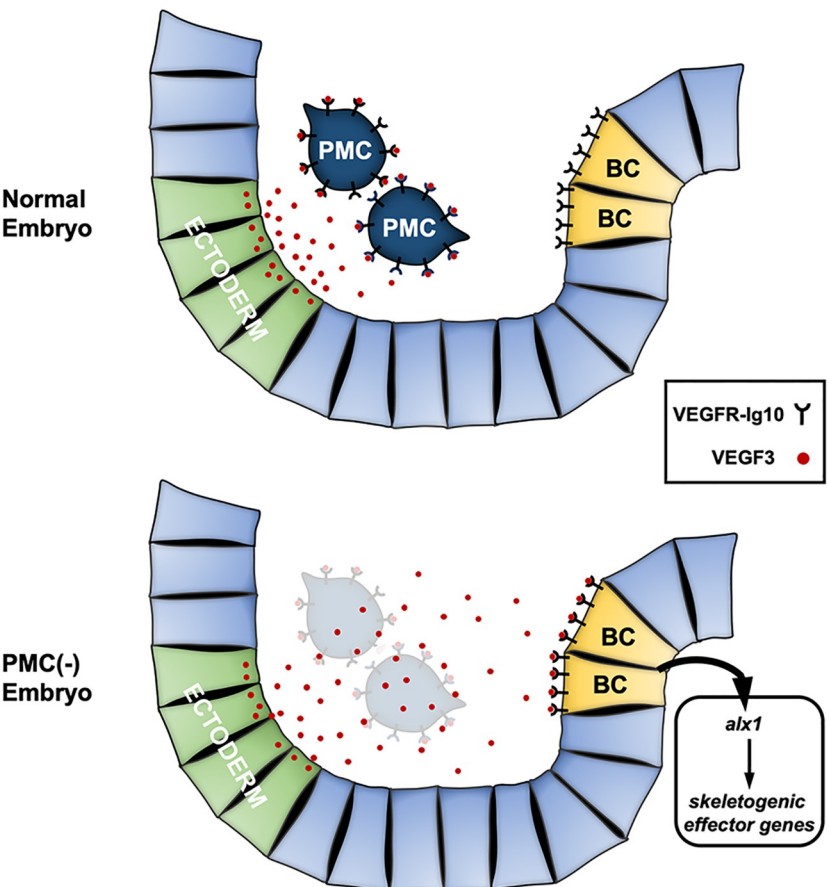

**Fig 9. A model of the PMC–BC interaction.** During normal development (top panel), PMCs migrate into the blastocoel at the onset of gastrulation, and VEGFR-Ig-10 on their surfaces sequesters VEGF3, which is expressed by ventrolateral ectoderm cells [9,11]. BCs are hypothesized to express low but functional levels of VEGFR-10-Ig. In PMC(−) embryos (bottom panel), VEGF3 is free to diffuse through the blastocoel and interacts with BCs. As a consequence of this signal, the key selector gene, *alx1*, is activated along with its many targets. At the same time, *alx1* suppresses competing regulatory states [24,29]. *alx1, aristaless-like homeobox 1*; BC, blastocoelar cell; PMC, primary mesenchyme cell; VEGF, vascular endothelial growth factor; VEGFR, VEGF receptor.

What is the mechanism by which *vegfr-10-Ig* is induced in BCs following PMC ablation? One possibility is that there exists a second mechanism of PMC-to-BC communication, entirely distinct from the proposed competition for VEGF3, that underlies the induction of *vegfr-10-Ig* in BCs. A much simpler hypothesis, however, is that BCs ordinarily express low levels of VEGFR-10-Ig on their surface, consistent with our finding that these cells transiently express low levels of *vegfr-10-Ig* mRNA (S1 Fig). We hypothesize that signaling by VEGF3 activates a positive-feedback mechanism that rapidly elevates the expression of *vegfr-10-Ig*. Notably, a similar mechanism normally operates in PMCs, in which *vegfr-10-Ig* expression is maintained by signaling through VEGF3 and VEGFR-10-Ig [9,11]. Robust activation of *vegfr-10-Ig* occurs even in the presence of emetine, arguing against any mechanism that requires the expression of new transcriptional regulators and instead suggesting that posttranslational modification of transcription factors that directly regulate *vegfr-10-Ig* might be responsible. Positive-feedback regulation of VEGFR expression by VEGF signaling has also been described in mammalian cells and has been attributed to the posttranscriptional regulation of several transcription factors that control VEGFR expression [46–48].

An important, unresolved issue is the mechanism by which VEGF signaling activates *alx1* in transfating BCs. This may be related to the mechanism by which VEGF3 ordinarily regulates the expression of *alx1* (and many other genes) within the PMC syncytium at late developmental stages, although the molecular basis of that control is also unknown. The effects of axitinib on PMC(−) embryos closely resemble those of U0126, an inhibitor of MEK, suggesting that VEGF might act through the MAPK cascade, as it does in other cell types [17,49]. One candidate mediator is E26 transformation-specific 1 (Ets1), an ERK-dependent transcription factor ordinarily expressed by both BCs and PMCs and a positive regulator of *alx1* [50].

The processes that have led to the appearance of new embryonic cell types are poorly understood [51–55]. Arendt and colleagues [54] recently formalized an evolution-based definition of cell type, which they consider to be a set of cells in an organism that change together through evolution, partially independent of other cells, and that are evolutionarily more closely related to each other than to other cells. According to this view, the key to the origin of a new cell type is gene regulatory independence—i.e., the capacity for regulating and evolving gene expression independently of other cells.

The micromere-PMC lineage is a recent evolutionary invention, and its appearance in euechinoids was associated with the coupling of an ancestral, adult skeletogenic program to the molecular polarity of the oocyte [4]. The intervening steps in the evolutionary appearance of PMCs are difficult to reconstruct, in part because of the great diversity of skeletogenic programs exhibited by modern echinoderms [3]. One hypothesis is that the ancestral echinoid program resembled that of modern cidaroids, the most basal group of living echinoids. These species have variable numbers of micromeres and lack an early-ingressing, skeletogenic mesenchyme; instead, a subpopulation of late-ingressing mesoderm cells produces the embryonic skeleton [56,57]. If this reflects the ancestral echinoid state, then the invention of PMCs may have been associated with mechanisms that suppressed the potential of late-ingressing, skeletogenic cells, thereby shunting them into alternative developmental pathways. It should also be noted that modern euechinoids possess non-micromere-derived, late-ingressing mesoderm cells that express a skeletogenic fate much later in development, after larval feeding begins [58]. This raises the possibility that BCs normally contribute to the late larval or adult skeleton in euechinoids and that PMC removal causes these cells to precociously activate the skeletogenic GRN.

In several respects, the gene regulatory programs of PMCs and BCs of modern euechinoids are similar. Many effector genes associated with a mesenchymal phenotype are coexpressed selectively by these two cell populations [59]. Both cell populations also express *ets1* and *ets-related gene* (*erg*) [60–62], two regulatory genes that provide positive inputs into *vegfr-10-Ig* in the PMC lineage [29]. The expression of these genes may reflect an ancestral, pan-mesodermal or pan-mesenchymal state, perhaps similar to that seen in the late-ingressing, nonskeletogenic mesenchyme of modern asteroids [63]. One striking exception to this shared pattern of gene expression, however, is the PMC-specific expression of *alx1*. This gene is activated specifically in the founder cells of the PMC lineage in the first cell cycle after their birth [28]. Alx1 has positive inputs into at least half of the >400 effector genes expressed selectively by PMCs and an even larger proportion of such genes that are expressed at high levels [24]. Recent chromatin immunoprecipitation sequencing (ChIP-seq) studies have shown that many of these inputs are direct [64]. *Alx1* also functions in PMCs to suppress alternative mesodermal regulatory states [24,29], and misexpression of *alx1* in nonskeletogenic mesoderm cells is sufficient to convert them to a PMC-like fate [17]. Moreover, the developmental expression and skeletogenic function of *alx1* are conserved across echinoderms [3]. Together, these findings highlight the pivotal role of Alx1 in specifying skeletogenic cell identity and establish this protein as a "terminal

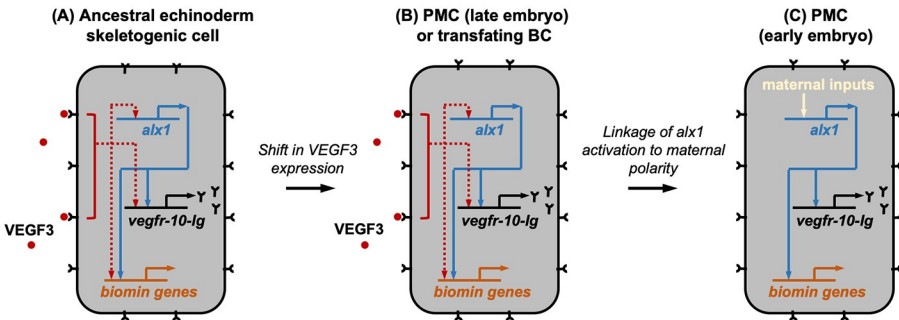

**Fig 10. A provisional model of skeletogenic cell type evolution in euechinoids.** (A) Ancestral skeletogenic cell, present in the adult of the last common ancestor of all echinoderms. Ancient roles for *alx1*, *vegf3*, and *vegfr-10-Ig* in echinoderm skeletogenesis are inferred from the conserved expression of these genes in adult and embryonic skeletogenic centers in multiple echinoderm clades and from experimental perturbations of gene function in echinoid and holothuroid embryos (see references in [3]). Red lines are inferred from experimentally determined inputs of VEGF3 signaling into *vegfr-10-Ig* and biomineralization genes in euechinoid PMCs [9,11,13]. The brokenness of the lines indicates that intervening transcription factors have not been identified. Regulation of *alx1* by VEGF signaling is hypothesized based on data presented in this study and from the restricted, ventral expression of *alx1* at the late gastrula stage in euechinoids [12]. Inputs from *alx1* into *vegfr-10-Ig* and other biomineralization genes (blue arrows) have been revealed by knockdown/overexpression of *alx1* in euechinoid and asteroid embryos [24,29,59,67]. (B) During echinoid evolution, the ancestral skeletogenic gene regulatory system was transferred into mesoderm-derived cells of the larva or late embryo. This may have only required a shift in *vegf3* expression in the ectoderm if a widely expressed receptor was linked to a feedback mechanism that up-regulated *vegfr-10-Ig*. In modern euechinoids, at postgastrula stages of development ("late embryo"), *alx1*, *vegfr-10-Ig*, and many biomineralization genes are regulated by VEGF signaling, which we suggest reflects the ancestral mode. Essentially the same regulatory machinery operates in BCs when VEGF3 is available, i.e., in PMC(−) embryos. (C) The evolution of PMCs involved the transfer of *alx1* expression into the large micromere lineage ("early embryo") by linking the activation of this gene to maternal β-catenin, its direct target, *pmar1*, and unequal cleavage [28,68,69]. The early, cell-autonomous activation of *alx1* in the large micromere-PMC lineage resulted in the precocious expression of VEGFR-10-Ig, which in modern euechinoids sequesters VEGF3 and isolates BCs from the exclusionary influence of Alx1, allowing these cells to express an alternative regulatory state. Alx1, Aristaless-like homeobox 1; BC, blastocoelar cell; *pmar1*, *paired-class micromere anti-repressor*; PMC, primary mesenchyme cell; VEGF, vascular endothelial growth factor; VEGFR, VEGF receptor.

selector" [54,65], i.e., a transcription factor that regulates a cell type–specific suite of effector genes while repressing alternative cell identities.

We propose that the evolutionary change that linked *alx1* activation to the prefertilization axis of the oocyte bypassed an ancestral mode of regulation that was based on VEGF signaling (Fig 10). The selective coexpression of *vegf3*, *vegfr-10-Ig*, and *alx1* in embryonic and adult skeletogenic centers in all echinoderms that have been examined strongly suggests that VEGF signaling had an ancient role in skeletogenesis and that expression of both *vegfr-Ig-10* and *alx1* was associated with the invention of skeletogenic cells very early in echinoderm evolution [4,6–8,66]. Although the evolutionarily derived, initial phase of *alx1* expression in euechinoids is independent of VEGF signaling [11], vestiges of an ancestral regulatory mechanism may persist in the apparent, signal-dependent regulation of *alx1* in the ventral domain of the PMC syncytium at the late gastrula stage [12]. We hypothesize that the VEGF-dependent expression of *alx1* in transfating BCs (see Figs 2C, 2C', 8B and 8B') is a manifestation of an ancestral regulatory mode. We further speculate that this mechanism may have involved feedback control whereby *alx1*, activated by VEGF3, up-regulated the expression of the cognate receptor, *vegfr-10-Ig*. An attractive feature of a feedback model is that the heterochronic transfer of the skeletogenic program would have only required a shift in the expression of the VEGF3 ligand, rather than separate, coordinated changes in the expression patterns of both the ligand and receptor, which are expressed in different germ layers. In the micromere-PMC lineage, although the ancestral mode of *alx1* activation has been superseded by new, maternally entrained mechanisms, *vegfr-10-Ig* remains a target of *alx1* [24,29,59]. One clear and

important outcome of the heterochronic shift of the skeletogenic program was the early, robust developmental expression of *vegfr-10-Ig* by PMCs. This expression begins at the blastula stage, well before PMC ingression [9,22]. Our findings indicate that the early expression of the receptor allowed PMCs to outcompete BCs for VEGF3, effectively isolating BCs from the powerful influence of *alx1* and allowing them to adopt an alternative regulatory state.

## Methods

### General methods

*L. variegatus* embryos were obtained and cultured as previously described [30]. Microsurgical removal of PMCs, PMC transplantation, vital labeling of embryos with RITC, and whole-mount immunostaining with mAb 6a9 were carried out according to Ettensohn and McClay [15]. WMISH using digoxigenin-labeled, antisense RNA probes was carried out as described previously [24]. Treatment with axitinib was carried out according to Adomako-Ankomah and Ettensohn [11].

### MO/mRNA injections

Microinjection of MOs and mRNA was carried out as described by Cheers and Ettensohn [70]. The sequence of the Lv-VEGFR-10-Ig splice-blocking MO was 5′-TGATTAGGGATTGC TACTTACCTGA-3′. The Lv-VEGF3 and Lv-IgTM MOs were described previously [11,37]. Working solutions (4 mM) of MOs were loaded into microinjection needles, and approximately 2 pl were injected into each egg. VEGF3 mRNA was synthesized as described by Duloquin and coworkers [9] and injected at 2–4 mg/ml (lower concentrations were also used for dose-response studies, as shown in Fig 8).

**Emetine treatments.** For emetine treatments, a stock solution of 100 mM emetine dihydrochloride hydrate (Sigma-Aldrich E2375) was prepared in deionized water and stored at −20 ˚C. Immediately before use, the stock solution was diluted 1:1,000 in ASW (final [emetine] = 100 μM). After embryos were loaded into a microsurgical chamber, the fluid in the chamber was replaced with 100 μM emetine, and microsurgery was carried out in the presence of the inhibitor. PMC(−) embryos were allowed to develop for an additional 2 hours in emetine-containing seawater before they were processed for WMISH.

**Injection of WGA into the blastocoel.** A 10 mg/ml solution of WGA (Sigma-Aldrich Cat. No. L9640) in seawater was prepared immediately before each use and front-loaded into microinjection pipettes. Control early mesenchyme blastula–stage embryos or PMC(−) embryos were immobilized in microinjection chambers [15], and sufficient WGA solution was injected into the blastocoel of each embryo to cause visible swelling. WGA-injected embryos were marked by co-injecting a droplet of silicon oil into the blastocoel. Embryos were fixed at various time points during gastrulation and immunostained with 6a9 antibody.

## Supporting information

**S1 Fig. Expression of *Lv-vegfr-Ig-10* at low levels in the presumptive nonskeletogenic mesoderm.** Whole-mount in situ hybridization analysis of *Lv-vegfr-10-Ig* expression in control, early gastrula-stage embryos. (A) Lateral view. (B) Vegetal view. Expression in the wall of the archenteron (presumptive nonskeletogenic mesoderm) is indicated by arrows. *Lv-vegfr-10-Ig, L. variegatus vascular endothelial growth factor receptor-10-Ig.*
(TIF)

**S2 Fig. BC transfating is not triggered by perturbing PMC migration.** (A) Control early gastrula-stage embryo. PMCs (arrow) have dispersed from the site of ingression and are migrating

along the blastocoel wall. (B) Sibling embryo, 4 hours after microinjection of WGA into the blastocoel. The PMCs (arrow) remain in a single mass at the site of ingression. The embryo is marked with an OD. (C) WGA-injected embryo 12 hours after injection, immunostained with 6a9 antibody. WGA has caused the coalescence of the PMCs into a single large mass from which numerous filopodia extend. No other 6a9(+) cells are present at the tip of the archenteron or in the blastocoel, indicating that BC transfating has not occurred. Analysis of WGA-injected, 6a9-stained embryos at multiple stages during gastrulation confirms that no BC cells transfate when PMC migration and patterning is disrupted. (D) PMC(−) embryo injected with WGA immediately after PMC removal and immunostained with antibody 6a9 after 12 hours. Numerous 6a9(+), transfated BCs are apparent, demonstrating that BCs are capable of transfating in the presence of WGA. BC, blastocoelar cell; OD, oil droplet; PMC, primary mesenchyme cell; WGA, wheat germ agglutinin.
(TIF)

**S1 Data. Counts of 6a9-positive cells.** Raw cell counts for all experiments presented in this paper. Each value represents the number of 6a9-positive cells in a single embryo. The data are shown graphically in the specific figure panels indicated.
(DOCX)

**S2 Data. Raw data underlying Fig 7.**
(XLSX)

## Acknowledgments

The authors thank Dr. Jennifer Guerrero-Santoro for her valuable contributions to this work.

## Author Contributions

**Conceptualization:** Charles A. Ettensohn.

**Data curation:** Charles A. Ettensohn.

**Formal analysis:** Charles A. Ettensohn.

**Funding acquisition:** Charles A. Ettensohn.

**Investigation:** Charles A. Ettensohn, Ashrifia Adomako-Ankomah.

**Methodology:** Charles A. Ettensohn.

**Project administration:** Charles A. Ettensohn.

**Resources:** Charles A. Ettensohn.

**Supervision:** Charles A. Ettensohn.

**Validation:** Charles A. Ettensohn.

**Visualization:** Charles A. Ettensohn.

**Writing – original draft:** Charles A. Ettensohn.

**Writing – review & editing:** Charles A. Ettensohn.

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
