## [Editor Report · Decision Letter 0]

5 Jun 2019

Dear Dr Ettensohn, 

Thank you for submitting your manuscript entitled "The evolution of a new cell type was associated with competition for a signaling ligand" for consideration as a Research Article by PLOS Biology.

Your manuscript has now been evaluated by the PLOS Biology editorial staff as well as by an academic editor with relevant expertise and I am writing to let you know that we would like to send your submission out for external peer review.

Please re-submit your manuscript within two working days, ie. by Jun 07 2019 11:59PM.

Kind regards,

Lauren A Richardson, Ph.D

Senior Editor

PLOS Biology

---

## [Decision Letter · Decision Letter 1]

2 Jul 2019

Dear Dr Ettensohn,

Thank you very much for submitting your manuscript "The evolution of a new cell type was associated with competition for a signaling ligand" for consideration as a Research Article at PLOS Biology. Your manuscript has been evaluated by the PLOS Biology editors, an Academic Editor with relevant expertise, and by several independent reviewers.

As you will read, the reviewers all appreciated the potential novelty and importance of your study. In light of the reviews (below), we will not be able to accept the current version of the manuscript, but we would welcome resubmission of a revised version that takes into account the reviewers' comments. We cannot make any decision about publication until we have seen the revised manuscript and your response to the reviewers' comments. Your revised manuscript may be sent for further evaluation by the reviewers.

Your revisions should address the specific points made by each reviewer. Please submit a file detailing your responses to the editorial requests and a point-by-point response to all of the reviewers' comments that indicates the changes you have made to the manuscript. In addition to a clean copy of the manuscript, please upload a 'track-changes' version of your manuscript that specifies the edits made. This should be uploaded as a "Related" file type. You should also cite any additional relevant literature that has been published since the original submission and mention any additional citations in your response. 

Before you revise your manuscript, please review the following PLOS policy and formatting requirements checklist PDF: http://journals.plos.org/plosbiology/s/file?id=9411/plos-biology-formatting-checklist.pdf. It is helpful if you format your revision according to our requirements - should your paper subsequently be accepted, this will save time at the acceptance stage.

Please note that as a condition of publication PLOS' data policy (http://journals.plos.org/plosbiology/s/data-availability) requires that you make available all data used to draw the conclusions arrived at in your manuscript. If you have not already done so, you must include any data used in your manuscript either in appropriate repositories, within the body of the manuscript, or as supporting information (N.B. this includes any numerical values that were used to generate graphs, histograms etc.). For an example see here: http://www.plosbiology.org/article/info%3Adoi%2F10.1371%2Fjournal.pbio.1001908#s5.

For manuscripts submitted on or after 1st July 2019, we require the original, uncropped and minimally adjusted images supporting all blot and gel results reported in an article's figures or Supporting Information files. We will require these files before a manuscript can be accepted so please prepare them now, if you have not already uploaded them. Please carefully read our guidelines for how to prepare and upload this data: https://journals.plos.org/plosbiology/s/figures#loc-blot-and-gel-reporting-requirements.

Upon resubmission, the editors will assess your revision and if the editors and Academic Editor feel that the revised manuscript remains appropriate for the journal, we will send the manuscript for re-review. We aim to consult the same Academic Editor and reviewers for revised manuscripts but may consult others if needed.

We expect to receive your revised manuscript within two months. Please email us (plosbiology@plos.org) to discuss this if you have any questions or concerns, or would like to request an extension. At this stage, your manuscript remains formally under active consideration at our journal; please notify us by email if you do not wish to submit a revision and instead wish to pursue publication elsewhere, so that we may end consideration of the manuscript at PLOS Biology.

When you are ready to submit a revised version of your manuscript, please go to https://www.editorialmanager.com/pbiology/ and log in as an Author. Click the link labelled 'Submissions Needing Revision' where you will find your submission record. 

Sincerely,

Lauren A Richardson, Ph.D

Senior Editor

PLOS Biology

Reviews

Reviewer #1: 

Summary of the results - In this paper the authors study the molecular mechanisms underlying the change of cell fate from blastocoelar to skeletogenic in sea urchin embryos when the endogenous skeletogenic cells are removed. When the skeletogenic cells are removed, non-skeletogenic lineages change their fate and generate larval skeleton. In this paper, the authors show that in this condition, ectopic expression of skeletogenic regulatory and biomineralization genes as well as VEGFR-10-IG is observed in the non-skeletogenic mesodermal cells and in the endoderm. The authors show that the ectopic expression of the skeletogenic genes and the generation of larval skeleton in these conditions does not happen when VEGFR is inhibited by axitinib, and verify that the specific genes involved are VEGF3 and VEGFR-10-IG (using specific morpholinos). The ectopic expression of VEGFR is not reduced by blocking protein synthesis, which indicates that the protein responsible for the transcriptional activation of VEGFR in these cells was there before the skeletogenic cells were removed. The authors then suggest that the mechanism responsible for blocking the skeletogenic potential of the non-skeletogenic cells in normal embryos is the sequestering of VEGF3 signal by VEGFR-10-IG expressed in the skeletogenic cells. To test that directly they study the effect of returning different number of normal or VEGFR-morphants skeletogenic cells into embryos where the endogenous skeletogenic cells were removed (a very elegant and technically demanding experiment). They show that as the number of returned skeletogenic cells increases, the number of transfating non-skeletogenic cells decreases. When VEGFR-10-IG is knocked-down the returned skeletogenic cells are less potent in inhibiting non-skeletogenic cell transfating. They then show that overexpression of VEGF3 in the embryo is sufficient to activate two key skeletogenic transcription factors in the non-skeletogenic mesoderm and in the endoderm. They suggest that in the ancestral echinoderm skeletogenic cells both Alx1 and VEGFR were activated by VEGF signaling and this regulatory linkages are seen in the late embryo and in the transfating blastocoelar cells. They suggest that the early activation of Alx1 in the large micromeres of the sea urchin embryos had led to the early activation of the skeletogenic program and VEGFR in these cells and this blocked the skeletogenic potential of the non-skeletogenic mesodermal cells. 

Overal assessment - The experiments in this paper are convincing and elegant. The question of de-differentiation between closely related cell lineages is very interesting and relevant to many other systems where de-differentiation occurs, including induced stem cells and even cancer. The proposed mechanism is intriguing, beautifully addressed and would be relevant to these other systems, which makes this paper highly suitable to Plos Biology. There are a few gaps in the suggested model and in the proposed evolutionary scenario that should be corrected before the paper can be accepted. Additionally I think that the presentation should be made more accessible to the broad readership of Plos Biology, that is, provide more explanation and necessary background. 

Detailed comments from major to minor:

The paper provide convincing evidence for the necessity of VEGF-signaling and VEGFR-10-IG expression in the non-skeletogenic cells for these cells' de-differentiation. However, it is not clear how VEGF3 ligand is received by the non-skeletogenic cells as VEGFR-10-IG is expressed only in the skeletogenic cells in normal embryos. Furthermore, the fact that VEGFR-10-IG and Alx1 turn on in the non-skeletogenic mesoderm and endoderm in VEGF3 overexpression indicates that these cells are indeed able to receive the VEGF3 signal, but with which receptor? This point is mentioned in the discussion and the authors suggest that VEGFR-10-IG might be expressed in low levels in the non-skeletogenic cells below the detection level of WMISH. But I think more effort should be made to try and identify the missing receptor. There is a low maternal level of the VEGFR-10-IG receptor and there is another VEGFR receptor, VEGFR7, that is expressed in these embryos in low levels. The authors should try and see if they can detect VEGFR-10-IG in the non-skeletogenic cells either by overstaining WMISH or by antibody for the protein, if available. I also think that WMISH of VEGFR7 would be relevant. It could be that VEGFR7 is initiating the response to VEGF3 and then VEGFR-10-IG is activated and continue this response. Another possibility is that at high concentrations VEGF3 could be acting through another RTK receptor. All these possibilities should be mentioned and more effort should be put in studying the spatial expression of VEGFR-10-IG and VEFGR7 in the non-skeletogenic mesoderm and endodermal cells (WMISH). 

The authors identify the transfating cells as blastocoelar cells (BCs) while the activation of Alx1, VEGFR-10-IG and Tbr is broader – both blastocoelar and endodermal cells are turning on these genes when the skeletogenic cells are removed. In previous papers this group had shown that endodermal cells are also capable of transfating. I therefore do not understand why throughout the paper the authors refer only to the blastocoelar cells when the cells that ectopically express Alx1, VEGFR-10-IG and become skeletogenic include also endodermal cells and possibly other non-skeletogenic mesodermal fates. If there is a reason for that, the authors should explain it, otherwise, they should not refer to the transfating cells as BC only. 

In the evolutionary scenario (Fig. 10) the authors state that VEGF-signaling activates Alx1 in the late sea urchin embryo and they refer to their previous paper (Sun and Ettensohn 2014). As far as I could tell, the effect of VEGF-signaling on Alx1 was not studied in this paper, but this paper shows that VEGF-signaling does not affect other genes that are expressed in the same cells as Alx1 at the plutei stage. Furthermore, Alx1 and VEGFR spatial expression do not overlap at late skeletogenic stages – Alx1 is expressed in the dorsal PMCs and later in the cells at the edges of the body rod cells (Sun and Ettenshon 2014) while VEGFR is expressed at the cells at the edges anterolateral and post-oral rods (Adomako and Ettensogn 2013). Additionally, a recent paper had shown that Alx1 is not affected by VEGFR inhibition up to the late gastrula stage (Morgulis et al 2019). I therefore believe that the statement that VEGF-signaling controls Alx1 expression at the late sea urchin embryo is not correct and should be removed from the paper. 

The common transcription factors expressed in both skeletogenic mesoderm (SM) and non-skeletogenic mesodermal (NSM) cells could be a key to the potential of the NSM to become SMs, and I think that more details should be provided about it in the manuscript. The upstream transcription factors that are common to the SM and NSM cells are, Ets1, Hex, Erg and Tel, three of which are Ets factors that can mediate VEGF-signaling. At early stages, Ets1 and Hex were shown to activate VEGFR expression in the skeletogenic cells together with Dri and Alx1 (Oliveri, Tu and Davidson, 2008). Dri is not an essential activator as it turns off at the skeletogenic cells while VEGFR expression lingers. Thus, is probably suffice to have Alx1 expressed in these cells to get VEGFR upregulated. I think that the similarity in the regulatory states between the SM and the NSM should be explained more thoroughly both in the introduction and discussion, as this is highly relevant to why these cells and not other cells are transfating. 

I think it would be interesting to discuss the findings of this paper in the light of two recent publications that studied the role of VEGF in sea urchin skeletogenesis (Morgulis et al 2019) and in brittle star skeletogenesis, where FGF signaling seem to have taken a lead role instead of VEGF signaling (https://www.biorxiv.org/content/10.1101/632968v1). Specifically, in the last paragraph of the paper that discusses the role of VEGF in skeletogenesis in echinoderms. 

The experiment that shows that VEGFR-10-IG turns on when protein translation is blocked is very interesting (Fig. 4). I wonder if they did the same experiment with Alx1 – does the initial transcriptional response mediated by proteins that are already present in these cells? 

The experiment where the authors return PMCs to the skeletogenic-depleted embryos is beautiful (Fig. 7), but I don't understand how come the fluorescence dye from the Donor PMC doesn't diffuse to the transfated cells in the recipient embryo. As far as I understand, all the skeletogenic cells are fused to each other and therefore dyes diffuse between them – but I could be wrong for donor cells. If this doesn't happen between donor and recipient cells this should be explained in the text. 

The experiment where the authors injected WGA to the embryos to show that VEGFR phenotypes are not due to defects in migration is only described in words. Representative images of this experiment would make it more visual and understandable, maybe as a sup figure.

The first paragraph of the discussion starts with a very general statement and the main point is only presented in the sentence before the last (the sequestering mechanisms). I think this paragraph should start with a stronger, more focused statement about sequestering mechanisms, and then define the gap in knowledge that this paper addressed. I think it would lead better to the rest of the discussion. 

The methods section is very brief and a description of the emetine protocol is missing.

In the second paragraph of the introduction in the third line the word "also" appears twice.

It would be helpful to add line numbers to the text.

---------------

Reviewer #2: Hiroshi Wada, signed review

This article provided an important findings on the mechanism of BC cell fate decision in sea urchin. The authors found that BC transfating requires VEGF signaling (mediated by VEGF3 and VEGF-10-Ig), and the VEGF signaling is required for the activation and maintenance.　Together with the VEGF overexpression and experimental perturbation of VEGFRs on PMCs, the authors concluded that VEGF sequestration is the likely mechanism for PMCs to inhibit skeletogenesis on BCs. 

I agreed that the evidences presented here are consistent with the ligand competition model of PMC vs BC skeletogenesis. But the evidence presented here may not strong enough to exclude the old hypothesis that PMCs send inhibitory signal. 

First of all, I suggest the authors to describe the alternative hypothesis with which they compare with the ligand competition model. Without that I felt it quite hard to get the point of the section “VEGF signaling is required for...” and “VEGF signaling acts through VEGFR-10...”. Perhaps the alternative hypothesis is that PMCs send some inhibitory signal to BCs. The two hypotheses may not be completely exclusive, but without that information, it was not easy for me to understand why the authors need to show VEGF signaling is required in BC transfating ( I felt that we already know that!). The most important difference between the old model and the model proposed here is in the presence or absence of inhibitory signal from PMC on BC transfating. Actually the PMC-specific expression of VEGF-10-Ig seems more consistent with the old hypothesis. In that sense, the most important finding is probably the VEGF inhibition suppress the Alx1 and Tbr expression upon PMC removal (Fig. 2C’, D’). But the data was obtained 4 hpd, although the authors show that Alx1 and Tbr are activated as early as 2 hpd. This observation is also consistent with the idea that, after the depletion of the inhibitory signal, Alx1 and Tbr activated at 2 hpd, but the maintenance of these expressions required vegf signaling. It would be critical to show whether the expression of Alx1 and Tbr in 2 hpd is affected by VEGF signaling inhibition. 

The authors say that the excess VEGF can induce BC transfating. But I have a concern on this conclusion. The experiment performed here was overexpression of ligand in the whole cells of embryos. Ligand was also expressed in BC cells, which was an unusual situation, and care need to be taken here. I suggest the author to design experiments in which high level expression of VEGF only in epidermal cells. 

Figure 9 is not accurate for the expression of VEGFR in BC cells. VEGFR is expressed at a very low level, if any, in BC cells. Isn’t it?

Regarding the issue on the novel cell type evolution, I suggest the authors to clarify whether the absence of specific expression of VEGFR in BC cells are the ancestral state of echinoderms before the acquisition of PMCs. Perhaps in the description of the ligand competition to be a driving force to generate novel cell types, some more explanation required to have the current status of PMCs and BCs in euechinoids. Or the VEGF ligand was competed by PMCs and BCs in both of which VEGFR show comparable levels of expression?

Implication from the observation that undetectable level of the receptors can sense the secreted ligand is very interesting from the aspect of evolutionary co-option of signaling system. I like that disucussion very much.

---------------

Reviewer #3: 

This manuscript makes an important contribution to the literature on the evolutionary origin of cell types. Most studies in this literature are purely descriptive, with the aims of establishing that a cell type is indeed novel and identifying the likely “parental” cell type from which it was derived. In this case, the origin of the novel cell type was previously identified by other groups and the authors of the present study seek a mechanistic understanding of how it originated. 

The experiments are thoughtfully designed, clearly described in the text, and well documented in the figures. The results presented support the direct conclusions that authors reach and provide persuasive evidence for their model (Figure 9). Overall, this is a beautifully conducted experimental study. The only technical concern I had was the lack of any real evidence to explain how the blastocoelar cells perceive VEGF3 (or the absence of PMCs if the signal is more indirect). The authors speculate that blastocoelar cells might produce a small amount of receptor that is simply too low to detect by in situs. Given that PMCs can be removed from live embryos, is it not possible to isolate a some blastocoelar cells and test this directly by qPCR?

A larger question that needs to be addressed more directly in the Discussion is why blastocoelar cells are still capable of responding to VEGF3 when it has apparently been hundreds of millions of years since they were the source for skeletogenic cells. It scarcely seems plausible that they are a kind of “wound healing” system given that the conversion to a skeletogenic fate happens upon removal of the PMCs. Surely this kind of thing only happens in the lab and not in real life? A possible explanation that the authors may have already considered is that the blastocoelar cells are actually the source population for the cells that make the adult skeleton and that the ability to differentiate into skeletogenic cells is part of normal development that simply takes place much later. Is there any evidence one way or the other to support this idea? It seems a bit dissatisfying to simply end by saying that the blastocoelar cells retain the ability to respond to VEFG3 and that’s just the way it is.

Minor points:

Use of the abbreviation hpd was confusing. At first I thought this was a typo. Since it is only used a few times, it would be clearer to spell it out.

A likely typo: second page of Results, line 4, should alx be alx1?

---------------

Reviewer #4: Günter P. Wagner, signed review

In this interesting and important paper the authors demonstrate an inhibitory interaction between primary mesenchymal cells (PMC) and blastocoelar cells (BC), and based on these results propose a scenario for the evolutionary origin of PMC, a derived cell type of the euechinoids. Specifically, the authors investigate the mechanisms through which BC can be induced to form skeletogenic cells, an activity that is normally suppressed by PMC. The experiments are well designed, executed and explained; alternative interpretations are considered and experimentally addressed.

This is an important paper since it is one of a very small number of experimental papers that directly address the mechanisms underlying the evolutionary origin of a new cell type. This is a revision of a previously submitted and reviewed (by other reviewers than this) paper and thus there is very little if anything that needs to be fixed. 

[EDITOR'S NOTE: this comment has been clarified with the reviewer and was due to a misunderstanding of our submission process and the R1 designation.]

 General questions and comments: 

From what is written in the manuscript it is not entirely clear why PMC are more capable of sequestrating VEGF3 than BC. Is it because PMC start expressing VEGFR earlier than BC can, and thus a critical part of the model is that PMC are earlier than BC and thus prevent the BC to receive a strong enough VEGF3 signal to activate the positive feedback between VEGF3 and VEGFR? 

Also, the evolutionary scenario, though very interesting and clear to follow, could be somewhat expanded. It seems that since BC can replace PMC in PMC(-) embryos, that these two cell types could be sister cell types. Is this supported by the phylogenetic distribution of larval/embryonic structures and phylogeny? Is the primary distinguishing characteristic of PMC that they are separate earlier from the blastoderm and then dominate the role of larval skeletogenic systems and the BC, though sister to PMC, evolved a new phenotype, that of a phagocyte? Or are the BC in other echinoderms both skeletogenic and phagocytotic? At least in asteroid larvae the mesenchymal cells are phagocytotic [this is the animal where macrophage like cells where discovered!] I think this paper will have a broader and deeper intellectual impact if the scenario in Figure 10 is a little bit expanded, including the information contained in the text about the skeletogenic cell types in cidaroids etc..

---

## [Editor Report · Decision Letter 2]

13 Aug 2019

Dear Dr Ettensohn,

Thank you for submitting your revised Research Article entitled "The evolution of a new cell type was associated with competition for a signaling ligand" for publication in PLOS Biology. 

The Academic Editor and I have now assessed your revision and we're delighted to let you know that we're now editorially satisfied with your manuscript. We will publish your manuscript, assuming you are willing to meet or final production requirements. Congratulations!

Before we can formally accept your paper and consider it "in press", we also need to ensure that your article conforms to our guidelines. A member of our team will be in touch shortly with a set of requests. As we can't proceed until these requirements are met, your swift response will help prevent delays to publication.

Please note that you may have the opportunity to make the peer review history publicly available. The record will include editor decision letters (with reviews) and your responses to reviewer comments. If eligible, we will contact you to opt in or out.

Sincerely,

Lauren A Richardson, Ph.D

Senior Editor

PLOS Biology

DATA POLICY:

Regardless of the method selected, please ensure that you provide the individual numerical values that underlie the summary data displayed in the following figure panels: (e.g. Figs. 2H, 3B, 5F, 6D, 7F, 8D), as they are essential for readers to assess your analysis and to reproduce it. Please also ensure that figure legends in your manuscript include information on where the underlying data can be found.

For manuscripts submitted on or after 1st July 2019, we require the original, uncropped and minimally adjusted images supporting all blot and gel results reported in an article's figures or Supporting Information files. We will require these files before a manuscript can be accepted so please prepare them now, if you have not already uploaded them. Please carefully read our guidelines for how to prepare and upload this data: https://journals.plos.org/plosbiology/s/figures#loc-blot-and-gel-reporting-requirements.

---

## [Editor Report · Decision Letter 3]

5 Sep 2019

Dear Dr Ettensohn,

On behalf of my colleagues and the Academic Editor, Marianne E. Bronner, I am pleased to inform you that we will be delighted to publish your Research Article in PLOS Biology. 

Early Version

PRESS 

Kind regards,

Hannah Harwood

Publication Assistant, 

PLOS Biology

on behalf of

Lauren Richardson,

Senior Editor

PLOS Biology